# Interaction between Cervical Microbiota and Host Gene Regulation in Caesarean Section Scar Diverticulum

AQ: au/A

Xing Yang,[a] Xinyi Pan,[a] Manchao Li,[a] Zhi Zeng,[a] Yanxian Guo,[a] Panyu Chen,[a] Xiaoyan Liang,[a] Peigen Chen,[a] Guihua Liu[a]

aReproductive Medicine Research Center, The Sixth Affiliated Hospital, Sun Yat-sen University, Guangzhou, Guangdong, People's Republic of China

**ABSTRACT** Cesarean section scar diverticulum (CSD) has become a formidable obstacle preventing women receiving CS from reproducing. However, the pathogenesis of CSD remains unexplored. In this study, we characterized the cervical microbiota, metabolome, and endometrial transcriptome of women with CSD. Based on the 16s rRNA results of cervical microbes, the microbial diversity in the CSD group was higher than that in the control group. *Lactobacillus* were significantly decreased in the CSD group and were mutually exclusive with potentially harmful species (*Sphingomonas, Sediminbacterium,* and *Ralstonia*) abnormally elevated in CSD. The microbiota in the CSD group exhibited low activity in carbohydrate metabolism and high activity in fatty acid metabolism, as confirmed by the metabolomic data. The metabolomic characterization identified 6,130 metabolites, of which 34 were significantly different between the two groups. In the CSD group, N-(3-hydroxy-eicosanoid)-homoserine lactone and Ternatin were significantly increased, in addition to the marked decrease in fatty acids due to high consumption. N-(3-hydroxy-eicosanoyl)-homoserine lactone is a regulator that promotes abnormal apoptosis in a variety of cells, including epithelial cells and vascular endothelial cells. This abnormal apoptosis of endometrial epithelial cells and neovascularization was also reflected in the transcriptome of the endometrium surrounding the CSD. In the endometrial transcriptome data, the upregulated genes in the CSD group were active in negatively regulating the proliferation of blood vessel endothelial cells, endothelial cells, and epithelial cells. This alteration in the host's endometrium is most likely influenced by the abnormal microbiota, which appears to be confirmed in the results by integrating host transcriptome and microbiome data. For the first time, this study described the abnormal activity characteristics of microbiota and the mechanism of host-microbiota interaction in CSD.

**IMPORTANCE** Cesarean section scar diverticulum (CSD) has become a formidable obstacle preventing women receiving CS from reproducing. In this study, we revealed that potentially harmful microbes do have adverse effects on the host endometrium. The mechanism of these adverse effects includes the inhibition of the activity of beneficial bacteria such as lactobacilli, consumption of protective metabolites of the endometrium, and also the production of harmful metabolites. In the present study, we elucidated the mechanism from the perspectives of microbial, metabolic, and host responses, providing an important rationale to design preventive and therapeutic strategies for CSD.

**KEYWORDS** cesarean section scar diverticulum, gene regulation, host-microbiota interaction, microbiome

Address correspondence to Guihua Liu, liuguihua@mail.sysu.edu.cn, Peigen Chen, chenpg@mail2.sysu.edu.cn, or Xiaoyan Liang, liangxy@mail.sysu.edu.cn.

The authors declare no conflict of interest.

Over recent decades, the cesarean section (CS) rate in China rose from 3% in 1988 to 34.9% in 2014, and then to 36.7% in 2018, ranking first among Asian countries (1, 2). Around 19.4% to 88% of women receiving CS will suffer from cesarean section scar diverticulum (CSD) (3). CSD is a result of poor healing of the local uterine incision, forming a depression or cavity that connects with the uterine cavity, which can result in a variety of near- and long-term complications, such as scar dehiscence, ectopic scar pregnancy, uterine rupture, prolonged menstrual bleeding, chronic pelvic pain, and secondary infertility (4, 5).

**TABLE 1** The clinical data features of 52 subjects

| n | CON 24 | CSD 28 | P | SMD |
|---|---|---|---|---|
| Age (mean [SD]) | 29.62 (3.57) | 33.25 (3.17) | <0.001 | 1.073 |
| BMI (mean [SD]) | S21.04 (2.25) | 22.23 (2.62) | 0.088 | 0.488 |
| AMH (ng/mL; mean [SD]) | 3.69 (2.09) | 3.42 (3.76) | 0.761 | 0.087 |
| Basal_FSH (IU; mean [SD]) | 6.99 (1.42) | 7.62 (1.91) | 0.192 | 0.372 |
| Basal_LH (IU; mean [SD]) | 5.51 (2.97) | 5.59 (3.38) | 0.929 | 0.025 |
| Basal_E2 (pg; mean [SD]) | 47.42 (68.52) | 42.89 (28.19) | 0.75 | 0.086 |
| Infertile_year | 3.54 (2.19) | 4.64 (2.91) | 0.134 | 0.428 |

CSD has become a formidable obstacle preventing those women from reproducing. The spontaneous pregnancy rate in CS women has decreased by 15%, and even with assisted reproductive technology, the embryo implantation and live birth rates are significantly lower (6). Our previous research revealed that persistent effusion is the major factor affecting fecundity in CSD women (7).

The female reproductive tract has a unique microbiome that has a critical role in the maintenance of homeostasis and/or development of certain diseases (8–10). When dysbiosis occurs, altered immune and metabolic signaling can produce commensurate responses, including chronic inflammation, epithelial barrier disruption, angiogenesis, and metabolic dysregulation (11–13). Our previous research revealed that persistent effusion in CSD were caused by localized inflammatory and immune imbalance, and local microbial disturbances play a core role in this process (14). However, the characteristics of local microbiota activities and the mechanism of microbial-host interaction are still unknown.

In this study, we combined nontargeted metabolomics and human host transcriptome to analyze the activity characteristics of CSD microbes and the mechanism of microbe-host interaction.

## RESULTS

**Diversity and compositional characteristics of cervical microbiota.** A total of 52 subjects were included in the study, including 28 in the CSD group and 24 in the CON group. The clinical data features of subjects are shown in Table 1. After filtering and quality control, an average of 60833.13 ± 7511.79 reads were obtained from each sample. The rarefaction curve indicated that the sequencing depth of the samples in this study was sufficient (Fig. S1A).

The $\alpha$-diversity of the microbiota calculated by Shannon-Wiener index at the phylum (Fig. 1A) and genus levels (Fig. 1B) in the CSD group was significantly higher than that in the CON group, indicating that the microbial composition of the CSD group was more abundant. Bray-Curtis-based PCoA plot and ANOSIM analysis showed that the distance between samples in the CSD group was significantly greater than in the CON group (Fig. 1C).

The absolute abundances of phylum (Fig. S1B) and genus (Fig. S1C) in the CSD group were higher than in the CON group. Classification at the phylum level showed a different pattern between the two groups, with *Firmicutes* being the overwhelming majority in the CON group, reaching 92% (Fig. 1D). In the CSD group, the proportion of *Firmicutes* decreased to 62%, and the proportion of *Proteobacteria* and *Actinobacteriota* increased to 20% and 10%, respectively (Fig. 1E). The composition of the two groups at the genus level also showed significant differences.

The present study dissected the genus composition of *Firmicutes* and *Proteobacteria* between the two groups. Under the *Firmicutes* genus, the proportion of *Lactobacillus* in the CSD group (Fig. 1F) decreased compared with that in the CON group (Fig. 1H) (84.91% versus 98.81%), while the proportions of *Streptococcus* and *Enterococcus* increased (7.62% and 3.33%, respectively). The proportions of Proteobacteria genera, including *Escherichia Shigella*, *Sphingomonas*, *Ralstonia* and *Burkholderia Caballeronia Paraburkholderia*, showed various degrees of differences between the CSD group (Fig. 1G) and the CON group (Fig. 1I). These results indicated that the diversity and composition of microbiota in the CSD group were

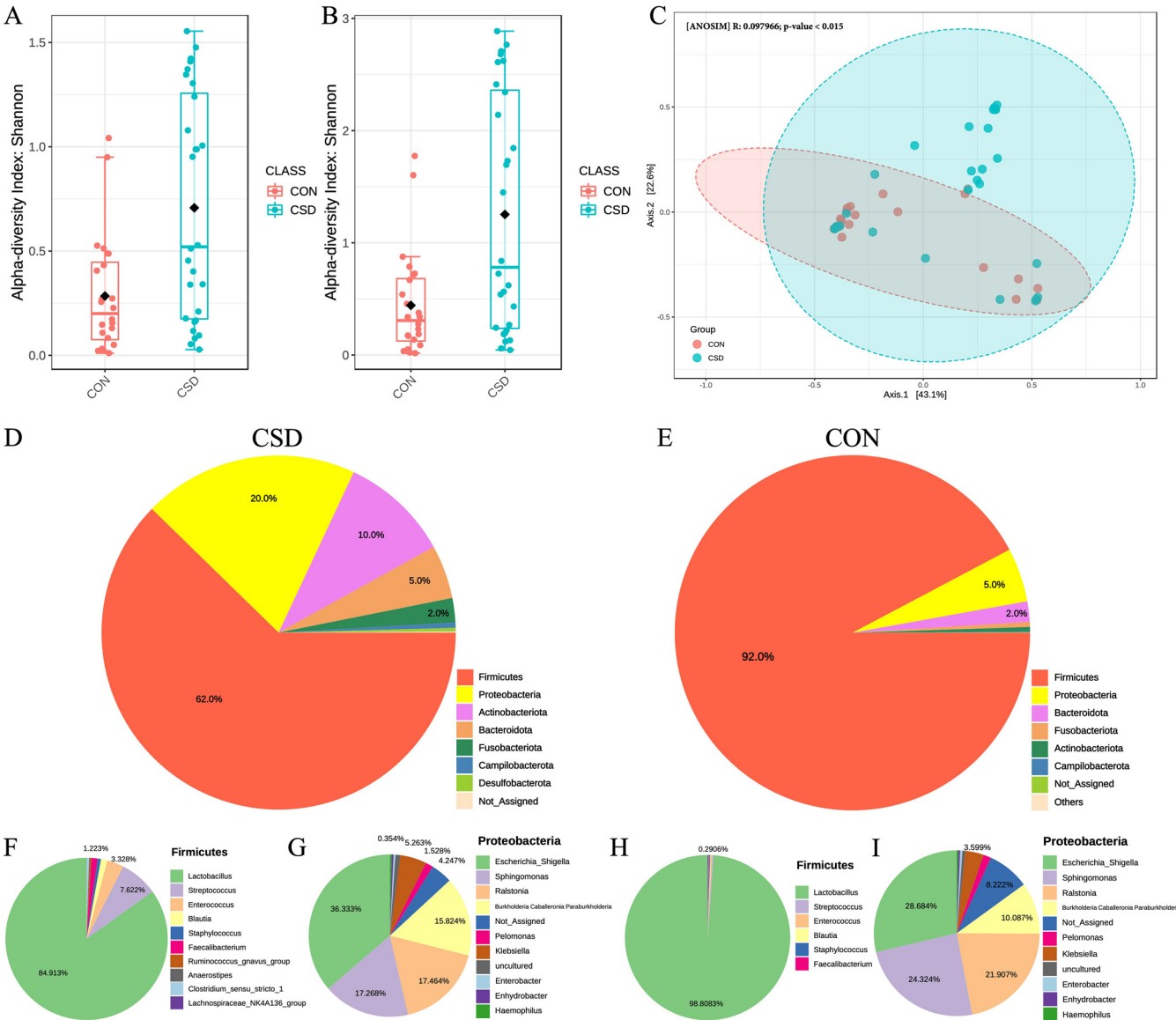

**FIG 1** Microbial community characteristics. The α-diversity of the microbiota calculated by Shannon-Wiener index at the phylum (A) and genus levels (B) (Wilcoxon rank sum test); (C) Bray-Curtis-based PCoA plot and Anosim analysis between CSD group and CON group; Pie chart of phylum composition of CSD group (D) and CON group (E); Pie chart of genus composition of *Firmicutes* (F) and *Proteobacteria* (G) in CSD group; Pie chart of genus composition of *Firmicutes* (H) and *Proteobacteria* (I) in CSD group.

significantly different from those in the CON group, and the proportion of *Lactobacilli* significantly decreased in the CSD group.

**Screening of differential microbiota and construction of cooccurrence network.** Based on linear discriminant analysis (LDA), differential genus between the two groups were screened. *Lactobacillus* significantly decreased in the CSD group, while *Gardnerella*, *Prevotella*, and other harmful genus increased significantly (LDA ≥ 2) (Fig. 2A, Table S1).

The genus cooccurrence network (Fig. 2B) constructed based on Spearman correlation analysis ($R > 0.8$, $P < 0.05$) showed interesting information. The cooccurrence network composed of *Ralstonia*, *Sphingomonas,* and *Sediminbacterium* was significantly negatively correlated with *Lactobacillus*. The abundance of these four genera was opposite between the two groups (Fig. 2C–F). This result suggested that the decrease in *Lactobacillus* abundance might be caused by the disturbance and mutual exclusion of the microbial community.

**Functional enrichment analysis of microbiota.** The results of PICRUSt 2 characterized the activity of microbiotas in two groups. We analyzed level 2 and level 3 of the KEGG

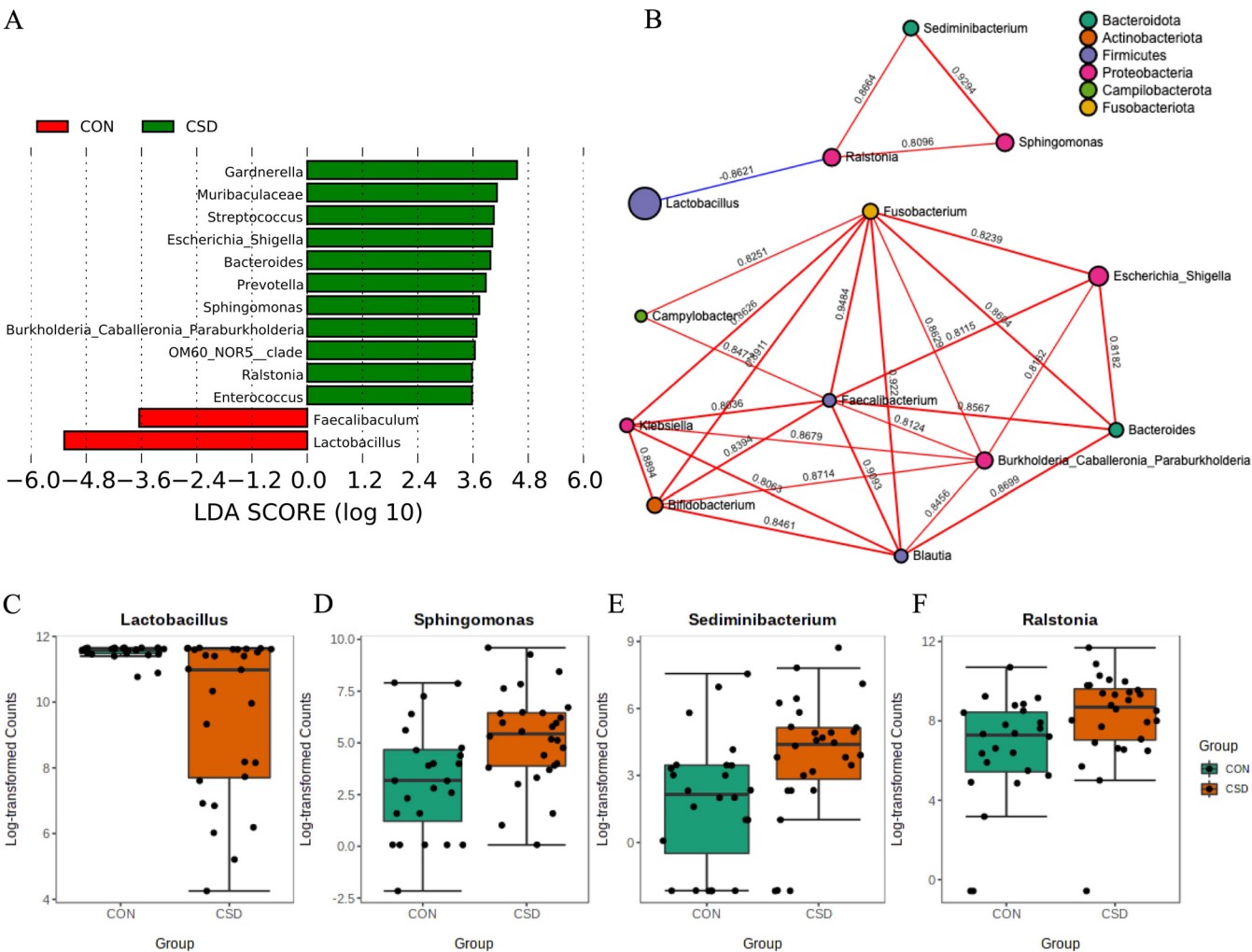

**FIG 2** Compositional differences and cooccurrence relationships of microbiota. (A) Differential genera between CSD group and CON group; (B) Cooccurrence networks of microbial communities (Spearman correlation analysis). The blue line indicates a negative correlation, and the red line indicates a positive correlation. The data on the line is the correlation coefficient. The size of the dots indicates the abundance of the genus; The abundance of *Lactobacillus* (C), *Sphingomonas* (D), *Sediminbacterium* (E), and *Ralstonia* (F) in CSD group and CON group (Wilcoxon rank sum test).

pathway, respectively. The CSD group showed significant activity in multiple metabolic processes, except for carbohydrate metabolism, an important pathway for lactate production (Fig. 3A). In a more refined pathway (level 3, Fig. 3B), the CSD group showed high activity in the fatty acid metabolism pathway and the biosynthesis of secondary metabolites, whereas the active carbohydrate metabolism pathway in the CON group was refined to glucose metabolism. Phosphotransferase system (PTS) and Fructose & mannose metabolism in the CON group were also more active than in the CSD group (Fig. 3B). The *P*-values were corrected using the BH (Benjamini and Hochberg) method.

**Nontargeted metabolomics in the cervical environment.** We performed simultaneous nontargeted metabolomic assays on samples from 60 subjects. After quality control (Fig. S2), 46 microbiome-matched samples were included in the subsequent analysis, including 20 samples from the CON group and 26 samples from the CSD group. After noise removal, 10,119 anion peaks and 8,308 cation peaks were obtained. The ion peaks with all missing values in the group (0 value) > 50% were deleted, and we replaced the 0 value with half of the minimum value and deleted the qualitative result score less than 36 points. As a result, we obtained 6,130 metabolites.

OPLS-DA was used to discriminate overall differences in metabolic profiles and further identify differential metabolites between groups. The results indicated that the samples

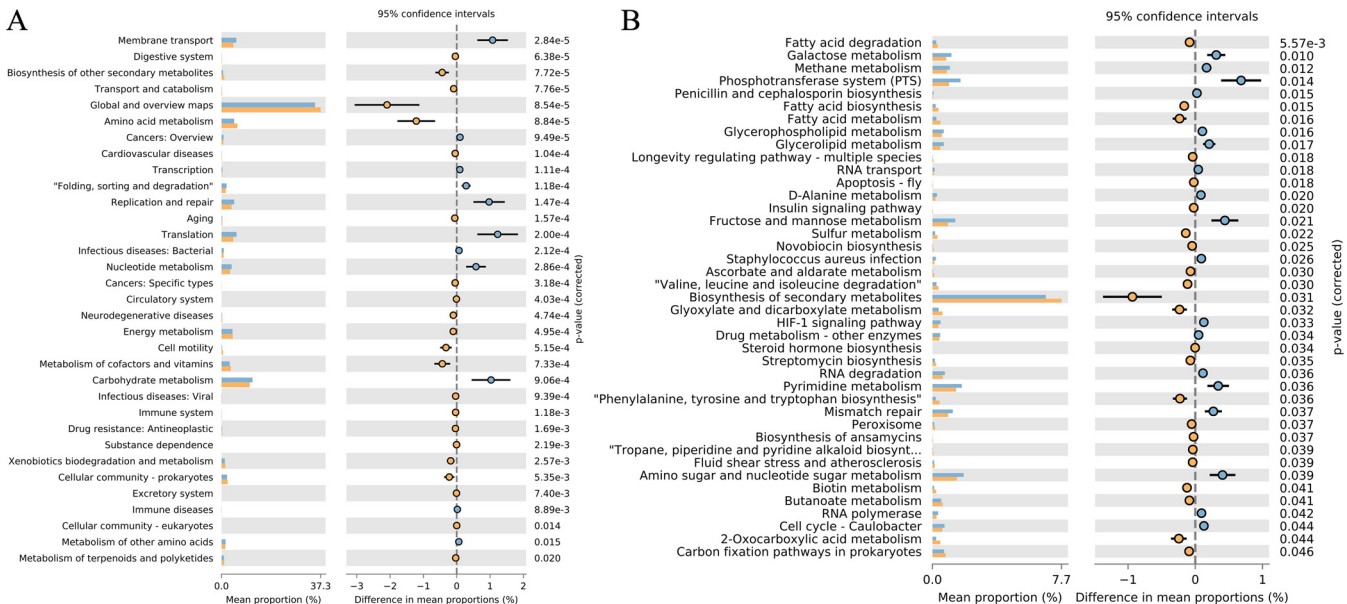

**FIG 3** Functional characterization of cervical microbes. Difference between CSD group and CON group in KEGG level 2 (A) and level 3 (B) (Welch's *t* test).

within the CSD group were clustered together and differed from the overall profile of the CON group (Fig. 4A, B). Thirty-four metabolites that differed significantly between the two groups were identified (Fig. 4C, Table S2). Two metabolites were significantly upregulated in the CSD group; namely, N-(3-hydroxy-eicosanoyl)-homoserine lactone and Ternatin. 32 metabolites were significantly downregulated, and the two most significantly downregulated metabolites were Gingerol and PC(O-10:0/O-8:0)[U].

**Correlation between cervical microbiota and metabolome.** To further explore the pathogenic mechanisms of cervical microbes in CSD, we performed an integrative analysis of the cervical microbiome and metabolome (Fig. 5). The results showed a trend consistent with the cooccurrence network (Fig. 2B). We found that many metabolites that were positively correlated with *Lactobacillus* were negatively correlated with *Prevotella*, *Sphingomonas*, *Ralstonia*, etc. Two metabolites were significantly positively associated with Lactobacillus, including Antanapeptin C (*R* = 0.39) and 3-Epipapyriteric acid (*R* = 0.39). However, N-Acetyl-a-neuraminic acid, N-Acetyl-b-neuraminic acid, N-(3-hydroxy-eicosanoyl)-homoserine lactone, and Ternatin were negatively correlated with various genus mutually exclusive with *Lactobacillus*.

**Human host endometrial response.** To further understand the human host response to a disturbed microbial community, we performed transcriptome sequencing of endometrium surrounding the cesarean section scar diverticulum. We performed transcriptome analysis of 33 samples paired with the microbiome and metabolome, including 18 samples from the CSD group and 15 samples from the CON group. A total of 982 differentially expressed genes were identified between the two groups (Fig. 6A), including 176 genes that were upregulated in CSD, and 806 that were downregulated (Fig. 6B, Table S3). We noted that upregulated genes in the CSD group negatively regulated the proliferation of blood vessel endothelial cells, endothelial cells, and epithelial cells (Fig. 6C). At the same time, these genes were also active in the endothelial cell apoptotic process. This suggest that local angiogenesis was hindered in the CSD group. Downregulated genes were mainly concentrated in immune system-related processes (Fig. 6D).

The integration of microbes and transcriptomes was achieved by constructing the O2PLS model. After 10-fold cross-validation, the model building parameters were finally set as *n* = 5, nx = 3, ny = 1, and the R2X was 0.91 and R2Y was 0.90 (Fig. 7A). These two parameters indicate that the model is reliable. Fig. 7B shows the top 15 loading features of the two omics.

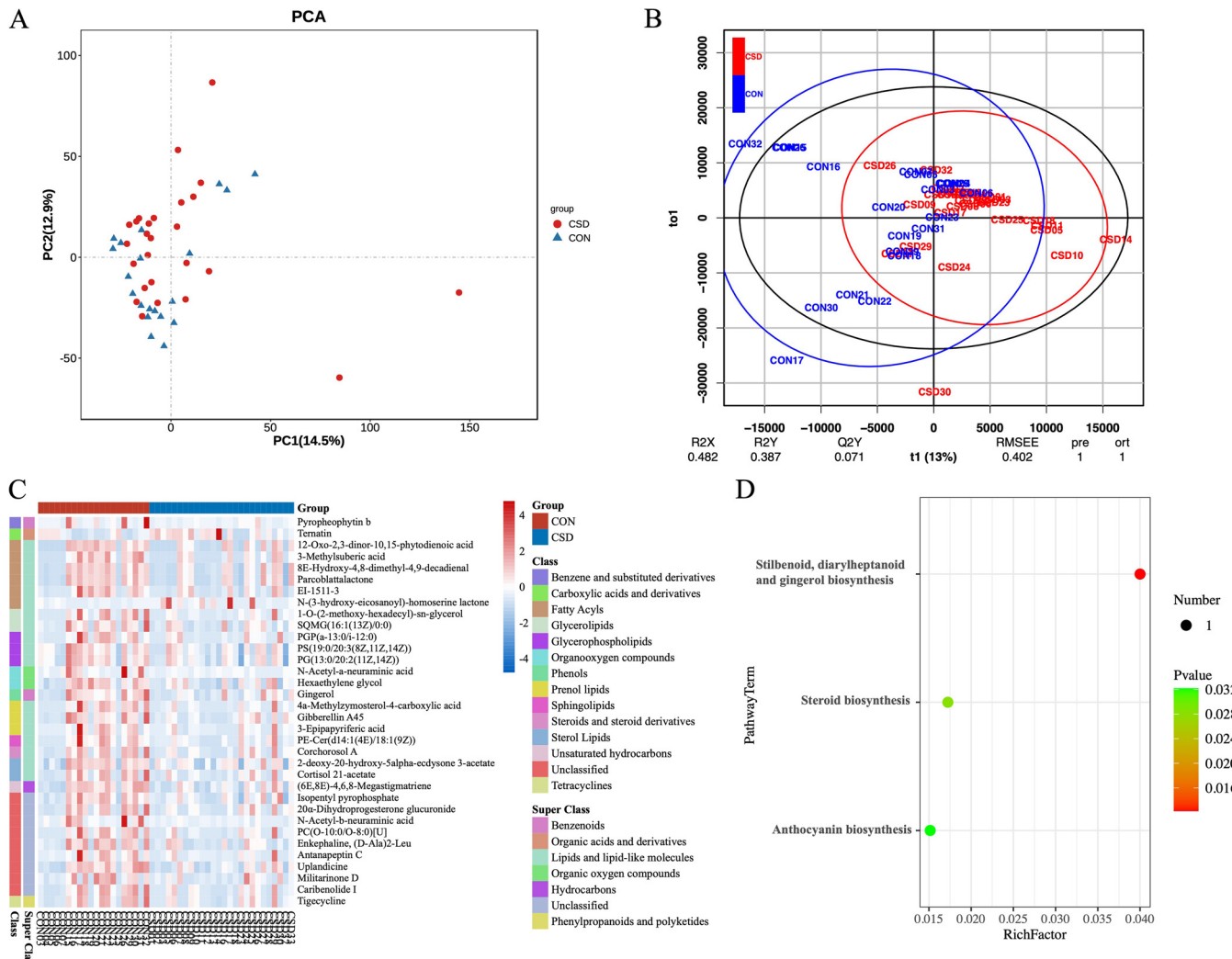

**FIG 4** Nontargeted metabolomics in the cervical environment. (A) PCA plots of the two groups of metabolites; (B) OPLS-DA score map. The abscissa represents the score value of the principal component, and the ordinate represents the score value of the orthogonal component; (C) Heatmap of differential metabolites between two groups; (D) KEGG enrichment analysis of differential metabolites.

We performed Spearman correlation analysis on the characteristics of top 30 loading in the two groups, trying to find out the relationship between specific genus and genes. We found several specific genes, including DKK1, CXCL14, SCARA5, APOD, S100A4, CFD, GPX3, and HBB. These genes were significantly positively correlated with genus mutually exclusive with *Lactobacillus* (Fig. 2B) but negatively correlated with *Lactobacillus* (Fig. 7C). The results of functional enrichment analysis of these genes showed that they were mainly active in the negative regulation of cell junction assembly and the process of epithelial to mesenchymal transition (Fig. 7D). Cell junction assembly was an essential process of angiogenesis. This result indicated that angiogenesis disorders and intimal hyperplasia disorders during the formation of CSD were closely related to microbial disorders.

## DISCUSSION

Cesarean section scar diverticulum (CSD) is a huge obstacle for those women who wish to have more children. This study attempted to explain the impact of cervical microbiota and metabolites on women with CSD from multiple perspectives, including the microbial perspective and the human host response perspective.

This study observed that the structure of cervical microbiota in the CSD group was different from that in the control group. *Lactobacillus* significantly decreased in CSD group,

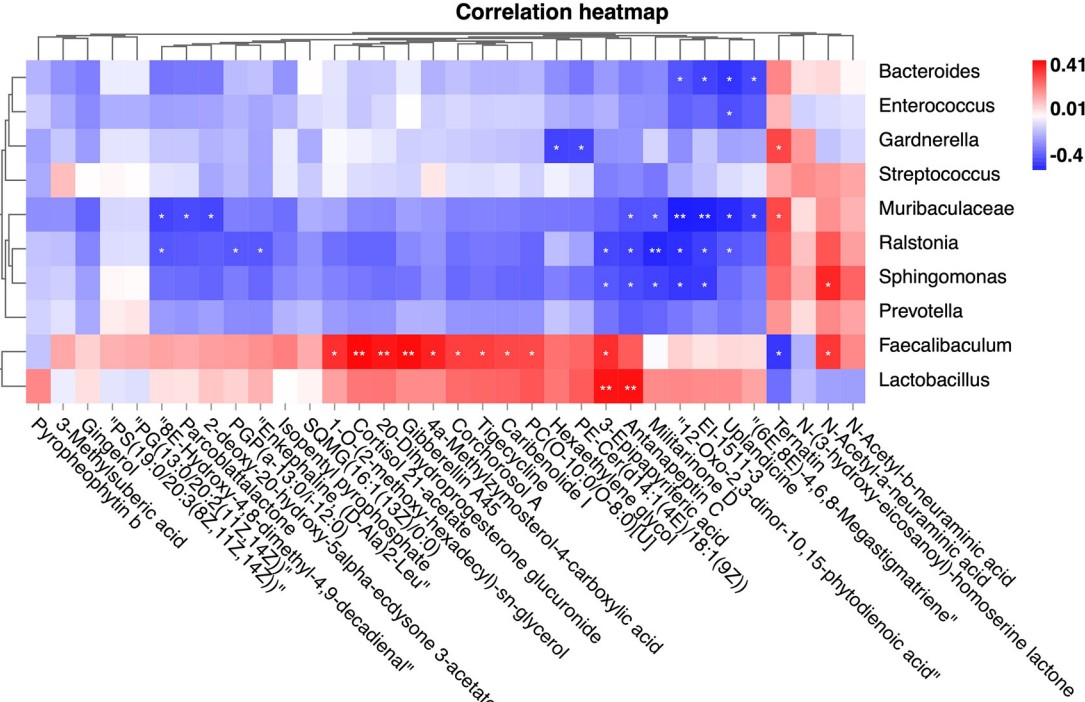

**FIG 5** Heatmap for the association analysis between differential metabolites and differential genus (Spearman correlation analysis).

while the proportion and abundance of potentially harmful species increased. The predominance of *Lactobacillus* was associated with vaginal health and the depletion of this genus can lead to many adverse conditions, such as an increased risk of acquired sexually transmitted infections (STIs), preterm birth, spontaneous abortion, or pelvic inflammatory disease (15). Lactobacillus maintains a healthy reproductive tract environment by producing lactic acid through its metabolic activity (16). In the cooccurrence network, we found a mutually exclusive relationship between *Lactobacillus* and three potential pathogenic bacteria (17, 18), including *Ralstonia, Sphingomonas,* and *Sediminbacterium* (here, referred to as the hostile genus of *Lactobacillus*). This was an interesting finding, and it was clearly very likely an important mechanism for the reduction of *Lactobacillus* in the CSD group.

In addition, we also observed that the CSD group significantly decreased the activity of the secondary pathway of carbohydrate metabolism and the tertiary pathway of galactose metabolism. The galactose metabolic pathway is an important pathway for *Lactobacillus* to produce lactic acid (19), and this result also confirmed that the activity of *Lactobacillus* was inhibited in the CSD group. Another active feature of the CSD group microbiota was the active metabolism of fatty acids. This result implied that fatty acids were largely consumed in the CSD group. Fatty acids played important roles in protection of the endometrium and multiple reproductive-related events, including gametogenesis, decidua, implantation, and placenta formation (20, 21). This finding was also confirmed in the metabolomic data of this study. Therefore, the inhibition of *Lactobacillus* metabolic activity and the active fatty acid metabolism of potentially pathogenic bacteria were important features of the CSD group, which might be important causes of adverse reproduction capacity.

To validate sequence-based characterization of microbial activity and to assess differences in cervical metabolites between the CSD and control groups, we identified 34 differential metabolites in the two groups. As mentioned above, the results of the metabolome were consistent with the sequence-based characterization of microbial activity inferred by the microbiome. In the CSD group, N-(3-hydroxy-eicosanoyl)-homoserine lactone and Ternatin were significantly increased, while various fatty acids were significantly decreased. Several of these fatty acids, which were decreased in the CSD group, were previously shown to

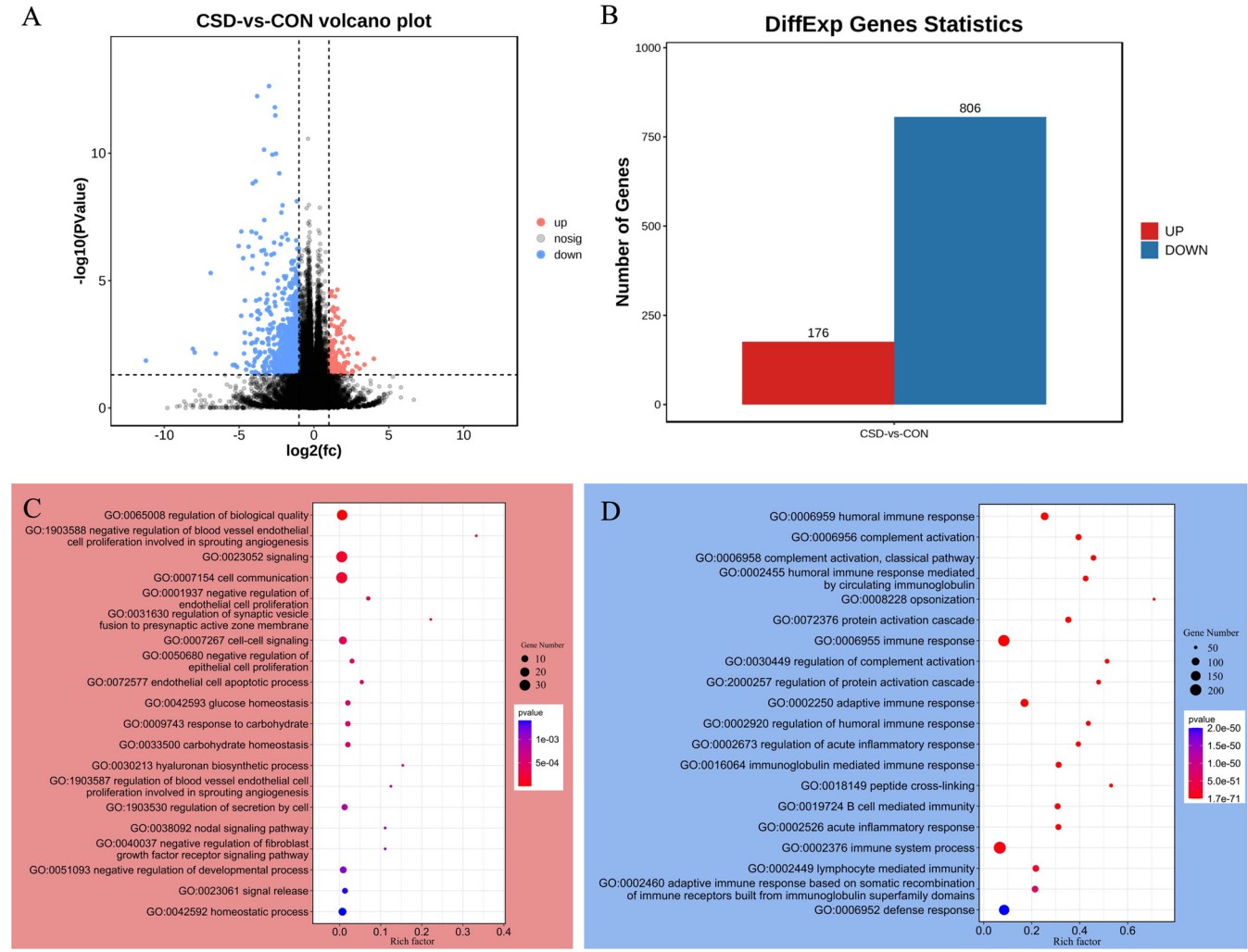

**FIG 6** Endometrial transcriptome expression characteristics. (A) Volcano plot for differential gene expression analysis; (B) Statistical chart of differentially expressed genes between CSD group and CON group; GO enrichment analysis of biological processes of upregulated (C) and downregulated (D) genes.

have protective effects on the endometrium, including 12-Oxo-2,3-dinor-10,15-phytodienoic acid, and 3-Methylsuberic acid (22). N-(3-hydroxy-eicosanoyl)-homoserine lactone has been shown to promote apoptosis of various cells *in vivo*, including macrophages (23), vascular endothelial cells (24), and fibroblasts (25), through mitochondrial damage received by reactive oxygen species. Interestingly, 12-Oxo-2,3-dinor-10,15-phytodienoic acid and N-(3-hydroxy-eicosanoyl)-homoserine lactone had the opposite relationship with the potentially harmful genus and *Lactobacillus*. The metabolomic results reconfirmed the mutually exclusive relationships in the cooccurrence network we found based on the microbiome data, namely, that the abundance and activity of *Lactobacillus* were significantly suppressed, and multiple beneficial fatty acids were largely consumed.

Although we had thoroughly explored the mechanisms and possible effects of microbial community disturbance in the CSD group, such as the discovery of potential pathogenic bacteria through the inhibition of *Lactobacillus* and the production of deleterious metabolism such as N-(3-hydroxy-eicosanoyl)-homoserine lactone, it is still unclear whether the human host was affected by these effects. Therefore, we performed transcriptome sequencing of the endometrium surrounding the CSD.

Surprisingly, we found changes in the transcriptome in the endometrium corresponding

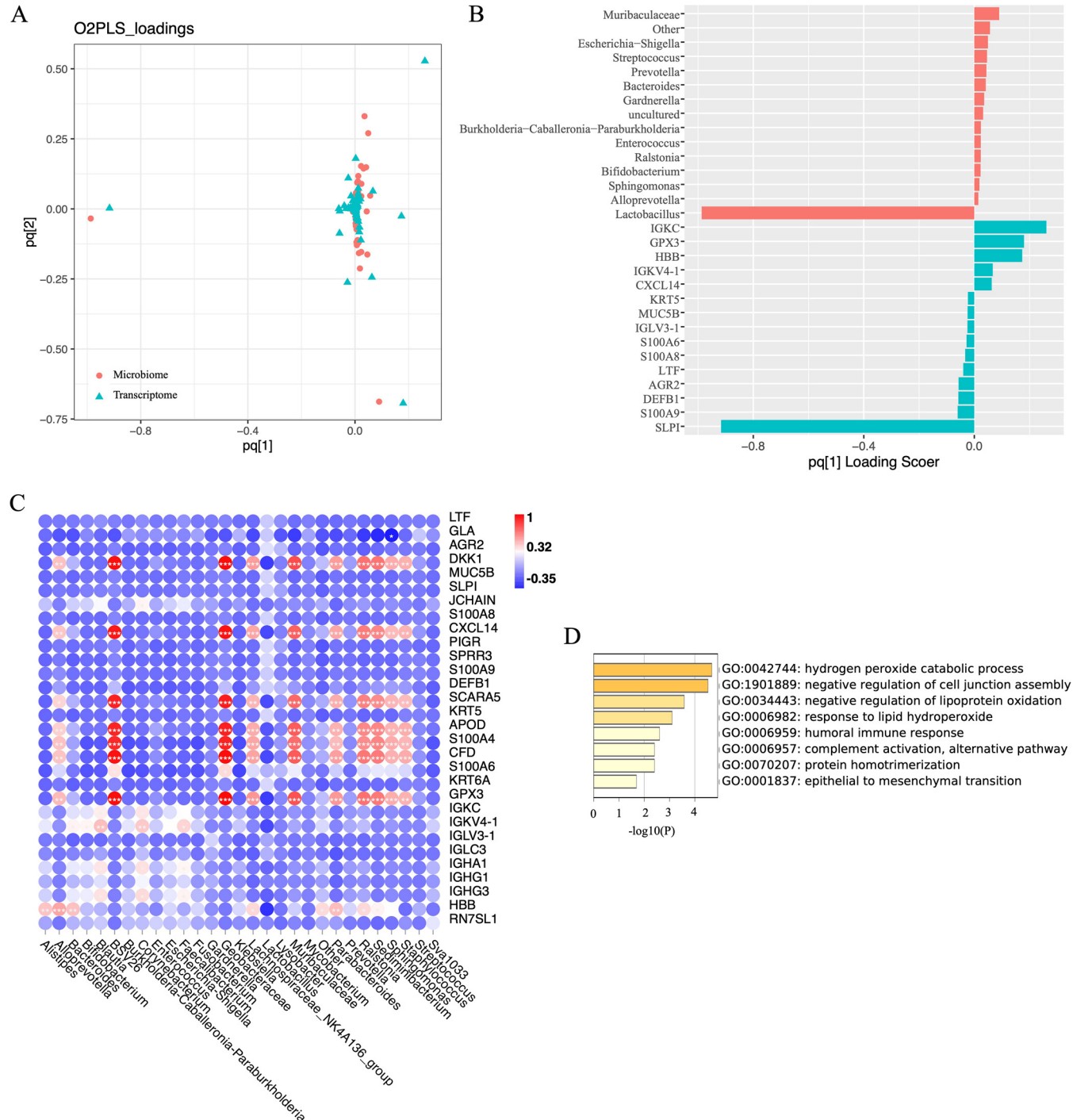

**FIG 7** Host-microbiota interaction analysis. (A) Loading scatterplot of the O2PLS model. The farther from the origin, the correlation of features is about stronger; (B) Bar graph of top 15 loading features in the microbiome and transcriptome. The characteristic trends in the same direction are consistent; (C) Heatmap of correlation analysis of top 30 loading features in both groups; (D) Functional enrichment analysis of Sphingomonas, Sediminbacterium, and Ralstonia-related genes.

to the above findings. GO enrichment analysis of the significantly upregulated genes in the CSD group showed that these genes were concentrated in the biological processes of negatively regulating the proliferation of blood vessel endothelial cells, endothelial cells, and epithelial cells. After integrating the microbiome and transcriptome data, we found that the genes regulating these functions were potentially pathogenic bacteria identified in the

previous data, including *Ralstonia*, *Sphingomonas*, and *Sediminbacterium*. *Lactobacillus* was negatively correlated with these adverse biological activities.

So far, we had revealed that potentially harmful microbes do have adverse effects on the host endometrium. The mechanism of these adverse effects includes the inhibition of the activity of beneficial bacteria such as *Lactobacilli*, consumption of protective metabolites of the endometrium, and also the production of harmful metabolites. However, the 16s rRNA sequencing method applied in this study has its limitations, such as short read lengths obtained and low species resolution (26, 27). In addition, the host's response requires more dimensions of experiments to aid validation. To our knowledge, this is the first study to detail the pathogenic mechanism of microbial community disturbance in CSD. Using the composition and activity characteristics of microbiota, ideas have been developed for the development of new and more effective probiotic formulations. At present, it has been reported that the *Lactobacillus rhamnosus* BPL005 strain can improve the health of the female reproductive tract (28). In addition, another study showed that among non-Lactobacillus-dominant patients treated with lactoferrin for 3 months after antibiotic treatment, 67% (6/9) of the patients had a return of the endometrial microbiota to a Lactobacillus-dominant environment (12, 29). These studies suggest that a combination of probiotics combined with prebiotics, such as lactoferrin, may have potential therapeutic benefits. An interventional study of *Lactobacillus* preparations is also under way, and we will announce the results of the study shortly. In the present study, we elucidated the mechanism from the perspectives of microbial, metabolic, and host responses, providing an important rationale to design preventive and therapeutic strategies for CSD.

**Conclusion.** Cervical microbiota in women with CSD had higher microbial diversity and lower *Lactobacillus* abundance. The cooccurrence network composed of *Ralstonia*, *Sphingomonas* and *Sediminbacterium* was mutually exclusive with *Lactobacillus* and inhibited local neovascularization and promoted apoptosis of vascular endothelial cells and endometrial epithelial cells by depleting protective fatty acids and producing N-(3-hydroxy-eicosanoyl)-homoserine lactone.

## MATERIALS AND METHODS

**Collection of research subjects and ethical approval.** Parous women underwent hysteroscopy at the Sixth Affiliated Hospital of Sun Yat-sen University in 2021 were enrolled in this case-control study. The inclusion criteria were (i) 20 to 40 years old; (ii) secondary infertility; (iii) normal karyotype; (iv) informed consent. The exclusion criteria included (i) acute pelvic inflammation, cervicitis, or vaginitis; (ii) endometriosis or adenomyosis; (iii) history of tuberculosis infection; (iv) antibiotics, glucocorticoids, or immunosuppressants received within a month before hysteroscopy; (v) sexual activity, vaginal irrigation, or drug application within 48 h before sampling. Parous women with post-cesarean section scar diverticulum (4, 5) were separated into the CSD group, and those who had vaginal deliveries were separated into the control group (CON group).

All study procedures were reviewed and approved by the ethics review board of the Sixth Affiliated Hospital of Sun Yat-Sen University (IRB no. 2019ZSLYEC-005S).

**Sample collection procession.** Cervix specimens were collected by using a sterile single-tipped CultureSwab. The swab was inserted into the cervix before hysteroscopy, rotated 360°, held for 10 s, removed and placed into the sterile collection tube, then stored at −80℃, avoiding vaginal walls contact. Uterine endometrium was taken from lower segment during hysteroscopy and immersed in 1 mL of RNAsafer reagent (R4811-02, Magen, China). Endometrial specimens were stored at 4℃ overnight before transfer to −80℃.

**16S rRNA extraction and sequencing.** The universal primers 341F (5′-CCTACGGGNGGCWGCAG-3′) and 805R (5′-GACTACHVGGGTATCTAATCC-3′) were used for 16s rRNA genes sequencing (30). DNA extraction and library construction were as described in our previous study (7). In short, according to the manufacturer's instructions, we used Magpure Soil DNA LQ kit (MAGEN, D6356-02) to conduct a total DNA extraction. PCR amplification was performed by Tks Gflex DNA polymerase (TaKaRa, R060B). After purification and quality control, NovaSeq 6000 (PE250) was used for high-throughput sequencing.

**Cervical microbiota analysis.** Raw data in the FASTQ format after Cutadapt software (31) was analyzed by QIIME2 (32). According to the default parameters of QIIME2, DADA2 (33) was used to perform quality filtering, noise reduction, splicing, and defitting and, finally, to obtain representative sequences and ASV abundance forms. Based on the Silva (Version138) (34) database, we used a q2-feature-classifier (35) to make a species comparison annotation. Features (ASVs) that count less than 1 in 10% of the sample were excluded.

The microbial diversity of cervical samples was estimated using Alpha diversity metrics inferred by the Shannon index (36). At the same time, the Wilcoxon rank-sum test was used to compare the diversity of the two groups. The Bray Curtis distance matrix was used in Bray Curtis principal

coordinates analysis (PCOA). Analysis of group Similarities (ANOSIM) was used to calculate statistical significance.

LEfSe (linear discriminant analysis effect size) (37) was used for differential analysis of cervical microbiota abundance.

We used Spearman correlation analysis to construct the correlation network between genus in Microbiome Analyst with $R > 0.8$ and $P < 0.05$ as cutoff values (38, 39).

PICRUSt2 (phylogenetic investigation of communities by reconstruction of unobserved states) (40) was used for predictive analysis of microbial function. Functional differences between groups were calculated by STAMP V2.1.3 (statistical analysis of taxonomic and functional profiles) (41).

**Nontargeted metabolomics.** We extracted metabolites from cervical swabs by adding 20 $\mu$L of separation buffer (methanol/acetonitrile/water [2:2:1]) to 1 mg of cervical secretions. A 10 $\mu$L isolate mix from all samples was set up as a quality control (QC) sample and used to assess stability during the experiment (42). 2 $\mu$L of isolate per sample was used to detect the signal of metabolites in all samples by liquid chromatography (LC) and mass spectrometry (MS) (ACQUITY UPLC I-Class plus, Waters). Raw data were qualitatively analyzed by metabolomics processing software Progenesis QI v2.3 with parameters of 5 ppm precursor tolerance, 10 ppm product tolerance, and 5% product ion threshold (Nonlinear Dynamics, Newcastle, UK). The internal standard normalization method was used for normalization of all output data, and the results were expressed as peak values (test sample peak area/internal standard sample peak area). Based on accurate mass-to-charge ratio (M/z), secondary fragmentation, and isotopic distributions, we used the Human Metabolome Database (HMDB), Lipidmaps (V2.3), Metlin, EMDB, PMDB, and self-built databases for compound identification.

The differential metabolites between the two groups were selected by the OPLS-DA (orthogonal partial least-squares discriminant analysis) method with VIP value of the first principal component $>1$, and the $P$-value value of the T-test $<0.05$ as cutoff. MetPA was used for differential metabolic pathway analysis (43).

In addition, Spearman correlation analysis was used to assess the correlation of microbiota with metabolites.

**Host RNA extraction and sequencing.** RNA extraction was performed using the RNeasy minikit (number 74104; Qiagen). Sequencing libraries were then generated using the NEBNext Ultra RNA Library Prep kit according to the manufacturer's instructions. Library preparations were sequenced on the NovaSeq 6000 (Illumina, Inc.) to generate 150 bp paired-end reads.

**Host RNA-seq analysis.** Trim Galore software was used for raw data quality control and junction trimming. We aligned the filtered raw data to the GRCh38 human genome using HISAT2 (44) software. The feature counts function (45) of the subread software (46) was used for gene quantification. The obtained count gene expression matrix was normalized to TPM (transcripts per kilobase of exon model per million mapped reads). The differentially expressed genes (DEGs) between the CSD group and CON group were selected by DEseq2 R package ($P$ value $< 0.05$ and log fold change $>1$) (47). Clusterprofiler R package (48) was used for functional enrichment analysis of DEGs.

**Host-microbiota interaction analysis.** In order to clarify the response mechanism of the human host's endometrium to microbial disturbances, the two-way orthogonal PLS (O2PLS) model was constructed for the integrated analysis of DEGs and genus using OmicsPLS R package (49). Spearman correlation analysis was used to determine the relationship between top30 loading DEGs and genus.

**Data availability.** The 16s rRNA gene sequencing have been deposited with China National Center for Bioinformation (https://ngdc.cncb.ac.cn/) under reference number PRJCA009374. The transcriptome sequencing for endometrium samples have been deposited with China National Center for Bioinformation under reference number PRJCA009373. Raw metabolome data are placed in Metabolights (http://www.ebi.ac.uk/metabolights/) under reference number MTBLS4967.

## SUPPLEMENTAL MATERIAL

Supplemental material is available online only.
**SUPPLEMENTAL FILE 1**, PDF file, 1.2 MB.
**SUPPLEMENTAL FILE 2**, XLSX file, 0.01 MB.
**SUPPLEMENTAL FILE 3**, XLSX file, 0.01 MB.
**SUPPLEMENTAL FILE 4**, XLSX file, 0.1 MB.

## ACKNOWLEDGMENTS

We thank the Shanghai Luming Biological Technology Co., Ltd. (Shanghai, China) for providing metabolomics services.

We declare that we have no conflicts of interest to disclose.

This study was supported by "Excellent Talents Training Project" of The Sixth Affiliated Hospital of Sun Yat-sen University (grant number R20210217202601970).

G.L., P.C., and X.Y. carried out the study. P.C. and X.Y. analyzed and interpreted the data and drafted the manuscript. X.P., M.L., Z.Z., and X.L. collected the samples. Y.G. and P.C. followed up and collected the clinical data. G.L., P.C., and X.Y. coordinated the study,

participated in the design, and reviewed the manuscript. All authors read and approved the final manuscript.

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
