## [Reviewer comments · Microbiology Spectrum]

Microbiology Spectrum

Interaction between cervical microbiota and host gene regulation in caesarean section scar diverticulum

Xing Yang, Xinyi Pan, Manchao Li, Zhi Zeng, Yanxian Guo, Panyu Chen, Xiaoyan Liang, peigen chen, and Guihua Liu

Corresponding Author(s): peigen chen, Sixth Affiliated Hospital of Sun Yat-Sen University

Review Timeline:

Submission Date:	May 8, 2022
Editorial Decision:	May 31, 2022
Revision Received:	June 3, 2022
Editorial Decision:	June 30, 2022
Revision Received:	July 1, 2022
Accepted:	July 1, 2022

Editor: Florence Doucet-Populaire

Reviewer(s): Disclosure of reviewer identity is with reference to reviewer comments included in decision letter(s). The following individuals involved in review of your submission have agreed to reveal their identity: Gajender Aleti (Reviewer #1); Din Ahmad Ud (Reviewer #2)

Transaction Report:

DOI: <https://doi.org/10.1128/spectrum.01676-22>

May 31, 2022

Dr. peigen chen
Sixth Affiliated Hospital of Sun Yat-Sen University
Reproductive Medicine Center
Guangdong 510000
China

Re: Spectrum01676-22 (Interaction between cervical microbiota and host gene regulation in caesarean section scar diverticulum)

Dear Dr. peigen chen:

Link Not Available

Sincerely,

Florence Doucet-Populaire

Journals Department
Reviewer comments:

Reviewer #1 (Comments for the Author):

As authors pointed out several metabolites identified in the CSD group are related to diet, could authors clarify if the participants dietary habits did not differ.

Could authors clarify how alpha-diversity was estimated? What was the minimum sequencing depth for data rarefaction? Authors should perform PERMANOVA analysis. Any confounding factors should also be included in the model. Should also test if the sampling groups compared have similar dispersion.

Authors should include a short description of FDR correction of all p-values if done.

Reviewer #2 (Comments for the Author):

Summary

The author investigated microbiome, Non-targeted metabolomics and transcriptomics data from control and CSD individuals. The author found a difference in diversity of microbiome, related metabolites etc in control and CSD group. Specifically Lactobacillus was decreased in CSD and comparatively in high amount in control individuals.

Following parts need to be revised.

1. I can't find this PRJCA009347, and reference number PRJCA009334 in that h China National Center for Bio-information. Maybe its not yet Published or any specific section ?
2. Figures numbers not shown/visible and so a bit difficult to find a specific figure easily; Only A,B,C D, are visible
3. Some figures quality need to improve to make able to see and read.
4. #Discussion Section. Line 275-276... Lactobacillus is italic and not in the following sentence. Also it does not mention that this Lactobacillus is high in which group; which author might intend control group; but seems missed to mention..
5. Most part of text sometime species/ genus names are italic and sometime not. Check and make corrections.
6. Line 313 mentioned lactobacillus and Lactobacillus two times?
- 7.Line 315 Lactobacillus not italic..
8. The author may include some recommendations lines in the last section of discussion; as the lactobacillus was diminished in CSD group and so, some probiotics containing lactobacillus maybe helpful in some way or the other in those individuals.

Staff Comments:

Preparing Revision Guidelines

Please return the manuscript within 60 days; if you cannot complete the modification within this time period, please contact me. If you do not wish to modify the manuscript and prefer to submit it to another journal, please notify me of your decision immediately so that the manuscript may be formally withdrawn from consideration by Microbiology Spectrum.

Reviewer #1:

Many thanks to the reviewers for giving us very pertinent comments. We have checked in detail point by point and made corrections.

- 1. As authors pointed out several metabolites identified in the CSD group are related to diet, could authors clarify if the participants dietary habits did not differ.**

Thanks for your valuable comments. We focused on the female reproductive tract metabolite status mainly related to the menstrual cycle, microbiota activity. We only focus on the part related to microorganisms in this study. The effect of diet on the microbial composition and activity of the female reproductive tract appears to be minimal. Previous studies have shown that the female reproductive tract microbiome is mainly affected by pregnancy status, menstrual cycle, sexual activity, age, and contraceptive use (DOI: 10.1007/s00203-021-02414-3, 10.1093/humupd/dmy012, 10.1002/jobm.202100421).

- 2. Could authors clarify how alpha-diversity was estimated? What was the minimum sequencing depth for data rarefaction?**

Thanks for your valuable comments. We supplemented the literature with information. The alpha-diversity of the microbiota was calculated by the Shannon-Wiener index. After filtering and quality control, an average of 60833.13 ± 7511.79 reads were obtained from each sample. The tag of each sample after rarefaction was 25796.

- 3. Authors should perform PERMANOVA analysis. Any confounding factors should also be included in the model. Should also test if the sampling groups compared have similar dispersion.**

Thanks for your valuable comments. We used ANOSIM (Analysis of similarities, a distribution-free similarity analysis) in our study. We have evaluated the test effect of PERMANOVA and found that ANOSIM seems to be more suitable for our study.

- 4. Authors should include a short description of FDR correction of all p-values if done.**

Thanks for your valuable comments. The p-values in the study were corrected

using the BH (Benjamini and Hochberg) method. We have supplemented the relevant information in the article.

Reviewer #2:

Many thanks to the reviewers for giving us very pertinent comments. The reviewer's comments have been checked and corrected point by point, and we have also checked the full text.

1. I can't find this PRJCA009347, and reference number PRJCA009334 in that h China National Center for Bio-information. Maybe its not yet Published or any specific section?

Thanks for your valuable comments. We have supplemented the detailed acquisition entry.

2. Figures numbers not shown/visible and so a bit difficult to find a specific figure easily; Only A, B, C, D, are visible

Thanks for your valuable comments. We have supplemented labelling of poorly annotated images (Supplement figure 2).

3. Some figures quality needs to improve to make able to see and read.

Thanks for your valuable comments. We double checked the uploaded image and replaced it with a higher resolution version. However, because the submission system compresses the image, the quality of the image may be degraded. This should not be the case at the time of official publication.

4. Discussion Section. Line 275-276... Lactobacillus is italic and not in the following sentence. Also, it does not mention that this Lactobacillus is high in which group; which author might intend control group; but seems missed to mention.

Thanks for your valuable comments. We have supplemented the original text. Corrected to “Lactobacillus significantly decreased in CSD group, while the proportion and abundance of potentially harmful species increased.”

5. Most part of text sometime species/ genus names are italic and sometime not. Check and make corrections.

Thanks for your valuable comments. We double checked and made corrections.

6. Line 313 mentioned lactobacillus and Lactobacillus two times?

Thanks for your valuable comments. It seems to be some typography issue. The former Lactobacillus is to describe the hostile genus of Lactobacillus, and the latter is Lactobacillus. To avoid ambiguity, we have rephrased it and corrected it in the original text.

7. Line 315 Lactobacillus not italic.

Thanks for your valuable comments. We have made corrections.

8. The author may include some recommendations lines in the last section of discussion; as the lactobacillus was diminished in CSD group and so, some probiotics containing lactobacillus maybe helpful in some way or the other in those individuals.

Thanks for your valuable comments. We add some of our suggestions in the last paragraph and highlight them.

June 30, 2022

Dr. peigen chen
Sixth Affiliated Hospital of Sun Yat-Sen University
Reproductive Medicine Center
Guangdong 510000
China

Re: Spectrum01676-22R1 (Interaction between cervical microbiota and host gene regulation in caesarean section scar diverticulum)

Dear Dr. peigen chen:

Link Not Available

Sincerely,

Florence Doucet-Populaire

Journals Department
Reviewer comments:

Reviewer #2 (Comments for the Author):

Following parts need to be adjusted.

1. Interestingly, these metabolites had the opposite relationship with the potentially harmful genus and Lactobacillus. Please Check this sentence after Ref 45.
2. The author didn't not included any references related to possible use of probiotics, there have a lot of probiotics which may enhance the diversity of those decreased tax, its not enough to say that "interventional study of Lactobacillus preparations is also underway", Where there are a lot of studies using and used various probiotics including lactobacillus already present which

can be cited to make this part strong in the recommendation part.

Staff Comments:

Preparing Revision Guidelines

Please return the manuscript within 60 days; if you cannot complete the modification within this time period, please contact me. If you do not wish to modify the manuscript and prefer to submit it to another journal, please notify me of your decision immediately so that the manuscript may be formally withdrawn from consideration by Microbiology Spectrum.

1. Interestingly, these metabolites had the opposite relationship with the potentially harmful genus and Lactobacillus. Please Check this sentence after Ref 45.

Thanks for your professional opinion. We have made changes in the original text after verification. Amended as "Interestingly, 12-Oxo-2,3-dinor-10,15-phytodienoic acid and N-(3-hydroxy-eicosanoyl)-homoserine lactone had the opposite relationship with the potentially harmful genus and Lactobacillus."

2. The author didn't not included any references related to possible use of probiotics, there have a lot of probiotics which may enhance the diversity of those decreased tax, its not enough to say that "interventional study of Lactobacillus preparations is also underway", Where there are a lot of studies using and used various probiotics including lactobacillus already present which can be cited to make this part strong in the recommendation part.

Thanks for your professional opinion. We make the necessary additional clarifications in the last paragraph. " At present, it has been reported that Lactobacillus rhamnosus BPL005 strain can improve the health of female reproductive tract (48). In addition, another study showed that among non-Lactobacillus-dominant patients treated with lactoferrin for three months after antibiotic treatment, 67 % (6/9) of the patients had a return of the endometrial microbiota to a Lactobacillus-dominant environment (12, 49). These studies suggest that a combination of probiotics combined with prebiotics, such as lactoferrin, may have potential therapeutic benefits. "

July 1, 2022

Dr. peigen chen
Sixth Affiliated Hospital of Sun Yat-Sen University
Reproductive Medicine Center
Guangdong 510000
China

Re: Spectrum01676-22R2 (Interaction between cervical microbiota and host gene regulation in caesarean section scar diverticulum)

Dear Dr. peigen chen:

Your manuscript has been accepted, and I am forwarding it to the ASM Journals Department for publication. You will be notified when your proofs are ready to be viewed.

Sincerely,

Florence Doucet-Populaire
Editor, Microbiology Spectrum

Journals Department
Supplemental table 2: Accept
Supplemental table 1: Accept
Supplemental Figure: Accept
Supplemental table 3: Accept